# The Effects of COVID-19 on Antifungal Prescribing in the UK—Lessons to Learn

**DOI:** 10.3390/jof10110787

**Published:** 2024-11-13

**Authors:** Katharine Pates, Zhendan Shang, Rebeka Jabbar, Darius Armstrong-James, Silke Schelenz, Jimstan Periselneris, Rossella Arcucci, Anand Shah

**Affiliations:** 1Department of Respiratory Medicine, King’s College Hospital NHS Foundation Trust, London SE5 9RS, UK; 2Department of Earth Science and Engineering, Imperial College London, London SW7 2AZ, UK; 3St Georges’ University of London, London SW17 0RE, UK; rebeka.jabbar1@nhs.net; 4Department of Infectious Disease, Imperial College London, London SW7 2AZ, UK; 5Royal Brompton and Harefield Hospitals, Guy’s and St. Thomas’ NHS Foundation Trust, London SW3 6NP, UK; 6Department of Microbiology, King’s College Hospital NHS Foundation Trust, London SE5 9RS, UK; 7Data Science Institute, Imperial College London, London SW7 2AZ, UK; 8Medical Research Council Centre of Global Infectious Disease Analysis, Department of Infectious Disease Epidemiology, School of Public Health, Imperial College London, London SW7 2AZ, UK

**Keywords:** antifungal, COVID-19, stewardship

## Abstract

Fungal infections are increasingly prevalent; however, antifungal stewardship attracts little funding or attention. Previous studies have shown that knowledge of guidelines and scientific evidence regarding antifungals is poor, leading to prescribing based on personal experiences and the inherent biases this entails. We carried out a retrospective study of inpatient antifungal usage at two major hospitals. We assessed the longitudinal trends in antifungal usage and the effect of COVID-19 on antifungal prescription, alongside levels of empirical and diagnostically targeted antifungal usage. Our results showed that the longitudinal patterns of total systemic antifungal usage within the trusts were similar to national prescribing trends; however, the composition of antifungals varied considerably, even when looking exclusively at the more homogenous group of COVID-19 patients. We showed a high level of empirical antifungal use in COVID-19 patients, with neither trust adhering to international recommendations and instead appearing to follow prior prescribing habits. This study highlights the significant challenges to optimise antifungal use with prescribing behaviour largely dictated by habit, a lack of adherence to guidelines, and high rates of empirical non-diagnostic-based prescribing. Further research and resources are required to understand the impact of antifungal stewardship on improving antifungal prescribing behaviours in this setting and the effects on outcome.

## 1. Introduction

Fungal infections are increasingly prevalent and account for more than 1.5 million deaths globally each year [1,2,3,4]. Treatment relies on a limited range of classes of systemic antifungal medications to which resistance is a rapidly growing problem [5,6,7]. A central principle for tackling antifungal resistance is the prevention of inappropriate and unnecessary use through stewardship programmes [8]. This may be at a national level, through standardisation of treatment practice and prophylaxis guidelines, or at a local level, through regular review of antimicrobial usage and auditing of prescribing practices. Historically, however, energy and resources in this area are directed towards antibiotic resistance, with little attention given to antifungals. Effective antifungal stewardship is further compounded by the challenges associated with diagnosing fungal disease, including non-specific signs and symptoms, low-sensitivity tests, and under-recognition by clinicians. Understanding the current trends of antifungal prescribing is crucial for designing and implementing targeted antifungal stewardship strategies.

In the UK, antifungal prescribing by National Health Service (NHS) Hospital Trusts remained relatively stable between 2016 and 2019; however, prescribing increased by 21% in 2020 [9]. This was attributed to the increased susceptibility of patients with SARS-CoV-2 (COVID-19) to COVID-19-associated pulmonary aspergillosis (CAPA). Although prescribing decreased slightly between 2020 and 2021, it remains above the pre-pandemic level [10]. The management of a single disease with susceptibility to fungal infection, however, enables the variability in antifungal prescription to be scrutinised.

In this retrospective cohort study, we analysed antifungal prescribing trends in two large NHS hospital trusts in South London with high pre-COVID-19 antifungal usage given the provision of tertiary cardiopulmonary, haematology, and hepatology services. We highlight longitudinal trends in antifungal usage and antifungal resistance and assess the effect of COVID-19 on antifungal prescription alongside levels of empirical and diagnostically targeted antifungal usage.

## 2. Materials and Methods

*Study type and data administration:* We conducted a retrospective study of inpatient antifungal usage by two large hospital trusts in South London: Kings College Hospital NHS Foundation Trust (KCH) and the Royal Brompton Hospital, part of Guy’s and St. Thomas’ NHS Foundation Trust (RBH). Data were extracted from electronic health records using the CogStack information and retrieval extraction platform [11]. Pharmacy dispensing records were used to obtain all inpatient prescriptions for the following antifungals: posaconazole, voriconazole, itraconazole, isavuconazole, caspofungin, micafungin, anidalufungin, and amphotericin B (any preparation). The location of prescription (intensive care unit or other ward) and COVID-19 status (positive, suspected, or negative) was collected. In patients with confirmed or suspected COVID-19, the following data were also collected: *Aspergillus* culture result (any source), blood fungal culture results (any species), and galactomannan test result (any source). All data were cleaned and processed using Python 3.8 software.

*Study period:* Data were collected from 1 January 2015 until 31 December 2021.

*Antifungal usage:* Usage was calculated as the number of distinct episodes of antifungal prescription. An episode was defined as a single course of an antifungal taken by a patient for longer than two consecutive days. If an individual was changed to a different antifungal, it was considered a new episode providing it continued for at least two consecutive days. Antifungal usage in the intensive care wards was calculated using the same methodology after adjusting for the location of the prescription.

*Antifungal usage in COVID-19 patients:* Overall antifungal usage in COVID-19 patients was calculated as the number of distinct episodes of antifungal prescription per month in patients with a confirmed or suspected COVID-19 admission (confirmed COVID-19 was determined by positive viral PCR; suspected was defined as individuals with clinical features typical of COVID-19, determined by a treating clinician within a COVID-19 pandemic wave). It was not possible to determine the temporal relationship of the antifungal prescription to the COVID-19 diagnosis; therefore, any antifungal episode within these patients’ admission episodes was counted. We then subdivided antifungal usage into diagnostic-based use or empirical use. The diagnosis of COVID-19-associated pulmonary aspergillosis requires a constellation of findings and has been separated into proven, probable, and possible based largely on biomarker or mycological evidence [12]. Given the difficulty of performing bronchoscopic sampling in the context of the COVID-19 pandemic, we considered patients to have diagnostic-based antifungal use if they met one or more of the following three criteria: positive *Aspergillus* culture (any source), positive fungal blood culture (any species), or galactomannan >0.5 nmol/mL (any source). A potential missed diagnosis included any patient who met any of the three diagnostic criteria described above but did not receive antifungal therapy, with empirical therapy defined as antifungal use without any of the criteria met.

*Aspergillus species resistance: Aspergillus* sp. resistance to any of the azole group (voriconazole, posaconazole, isavuconazole, and itraconazole) or amphotericin were collected. If there was resistance to more than one drug, it was counted towards each drug individually. Azole-resistant isolates were confirmed with a standard microbroth dilution method according to EUCAST reference guidelines [13].

*Ethics statement:* Retrospective electronic health record data collection and protocols were approved by the UK Research Ethics Committee (REC reference:21/HRA/3400)

## 3. Results

*Trust demographics:* KCH encompasses two hospitals: a large teaching hospital with tertiary hepatology and haematology services, and a district general hospital. RBH encompasses a single large hospital with tertiary cardiopulmonary services.

*Antifungal usage:* Total systemic antifungal usage at KCH increased by 13% between January 2015 and December 2021 (Figure 1A). Usage remained relatively stable between 2015 and 2020, before rising to a peak in 2021 within the COVID-19 pandemic. In contrast, peak antifungal usage at RBH was seen in 2019, followed by a 30% decrease by December 2021 over the duration of the COVID-19 pandemic (Figure 1B). Both trusts showed a small peak in 2017 followed by decreased usage in 2018.

The composition of antifungal usage, however, was notably different across the two trusts, which may likely relate to the differing populations served by the trusts (hepatology/haematology at KCH and cardiorespiratory at RBH). The echinocandin-class antifungal anidulafungin was the most used antifungal across all years at KCH, followed by posaconazole and amphotericin. In contrast, at RBH, amphotericin made up a much larger percentage of total usage, with echinocandin caspofungin the most commonly used echinocandin, and anidulafungin use extremely low. Use of voriconazole, which would commonly be the first-line antifungal guideline-based recommendation for invasive mould infection, was low at both sites [14].

*Antifungal usage in intensive care:* At KCH, total systemic antifungal usage in intensive care remained relatively stable between January 2015 and December 2019 (Figure 2A), following a similar pattern to overall usage. In contrast, however, there was a steep 90% increase in usage between 2019 and 2020, with a continued rise into 2021 during the COVID-19 pandemic. At RBH, the overall patten of systemic antifungal usage in the intensive care environment (Figure 2B) was very different to the total usage. In contrast to the decrease in total systemic antifungal usage seen between 2017 and 2021, usage doubled in the intensive care environment between 2017 and 2020, although there was a small decrease into 2021. Although both trusts showed the highest levels of usage in 2020 and 2021, at KCH this was a much larger increase from 2019 compared to RBH, where there was a steady increase in use across the preceding years.

Similar to total antifungal usage, the predominant antifungal used at KCH intensive care was anidulafungin, whereas RBH used almost exclusively amphotericin.

*Antifungal usage in COVID-19 patients:* KCH recorded 4144 unique patients with confirmed or suspected COVID-19 and RBH recorded 451 patients with confirmed or suspected COVID-19 in the time period analysed. Both trusts showed the highest usage of antifungals in COVID-19 patients in April 2020 and January 2021, mirroring waves of the COVID-19 pandemic (Figure 3A,B). Despite the homogenous disease population, however, the composition of antifungals used remained very different between the two trusts. The most utilised antifungal in COVID-19 patients at KCH was anidulafungin, followed by amphotericin, whereas RBH use was almost exclusively amphotericin.

At RBH, of the 451 patients with COVID-19, 265/451 (59%) received at least one course of systemic antifungals (Table 1). A total of 68 of these 265 (26%) were considered to have received antifungal therapy based on positive diagnostic tests. Of the 186 patients who did not receive antifungals, 18 (10%) would have met the diagnostic criteria. Of those who met the diagnostic criteria, 60/86 (70%) had a positive galactomannan test and 52/86 (60%) had relevant positive microbiology. In total, galactomannan testing was conducted on 79/451 patients (18%). It was not possible to ascertain how many patients had a fungal culture taken that showed no growth.

At KCH, of the 4144 patients with COVID-19, 171/4144 (4%) received at least one course of antifungals. Unfortunately, it was not possible to reliably ascertain galactomannan or culture results for KCH; therefore, these data were excluded from analysis.

*Aspergillus species resistance:* Lastly, to understand whether amphotericin use related to increasing azole resistance, we further analysed *Aspergillus* species resistance over time. Results were, however, only available for RBH. The total number of resistant *Aspergillus* cultures increased from 10 in 2015 to a peak of 30 in 2019, with a reduction during the 2020 and 2021 COVID-19 pandemic (Figure 4). The highest resistance was against itraconazole, peaking in 2019.

## 4. Discussion

In this study, we describe how systemic antifungal prescribing patterns differ in two geographically proximate hospitals both prior to and within the COVID-19 pandemic. Total consumption as well as the relative use of different antifungals varied considerably between the two trusts, even when looking exclusively at a homogenous susceptibility group of COVID-19 patients.

National data covering the same time period as our study show that the overall consumption of antifungals by NHS acute hospital trusts remained stable between 2016 and 2019, followed by a 21% increase in 2020 [9]. Consumption in 2021 was lower than in 2020, suggesting that this may have been the start of the return to pre-pandemic levels [10]. Between 2016 and 2019, our trusts did not differ much from national prescribing trends, with the slight peak in 2017 seen at both trusts possibly being due to the large *Candida auris* outbreak within London hospitals [15]. Between 2019 and 2020, in contrast to national data, our analysis shows an overall decrease in antifungal usage of 4% and 17% at KCH and RBH, respectively. This may be due to a high level of baseline use given the specialist services offered by both our trusts, which would likely have been suspended or greatly reduced at the height of the pandemic.

Interestingly, pre-COVID-19, the local antifungal prescribing patterns at the two trusts differed both from each other as well as from national trends [10]. Most notably, there was a much higher use of echinocandins, particularly anidulafungin at KCH and amphotericin at RBH. The increased pre-pandemic echinocandin use is perhaps partly explained by the specialist services offered by the trusts, with KCH having a large specialist hepatology unit. This would explain higher rates of invasive candidiasis, where echinocandins are the first-line recommended therapy and anidulafungin is the only member of the echinocandin group not metabolised by the liver [16,17]. However, the rationale for the high pre-pandemic use of amphotericin at both sites, particularly RBH, is not clear, as it is not an internationally recommended first-line antifungal therapy for most invasive fungal infections. Current international recommendations suggest the use of voriconazole for most cases of invasive aspergillosis [14,18] and an echinocandin for candidaemia [16]. As RBH is a centre providing extra-corporeal membrane oxygenation (ECMO), it is possible that the high use of amphotericin reflects a distrust of how ECMO affects voriconazole pharmacokinetics [19]. At KCH, it may be that azoles could not be used in their large haematology population due to drug–drug interactions. It is possible, however, that our findings reflect a lack of understanding of antifungal practice recommendations, which is consistent with multiple previous studies that show widespread poor antifungal prescribing practice amongst intensive care physicians [20,21,22,23]. Knowledge of guidelines and scientific evidence regarding antifungals is poor, leading to prescribing based on personal experiences and the inherent biases that this entails [24]. Previous work by Valerio et al. and Ibrahim et al. has shown that only 57% and 38% of physicians, respectively, knew that voriconazole was the best first-line therapy for invasive aspergillosis [20,24]. In addition, healthcare professionals may have limited experience with newer therapies such as posaconazole or isavuconazole, with improved pharmacokinetics, or may have had a bad experience with side effects or toxicity from a particular therapy—for example, hepatotoxicity with voriconazole—leading to reliance on more familiar antifungals (e.g., amphotericin).

The advent of the COVID-19 pandemic has enabled us to analyse antifungal prescribing within a homogeneous disease, with little difference in patient populations, especially given the proximity of the hospitals, and thus gain key insights into antifungal prescribing practices and behaviours. The possibility of CAPA was highlighted early in the context of the pandemic. International recommendations and guidelines suggest that voriconazole or isavuconazole should be considered the first-line agent, with amphotericin and posaconazole as second-line therapy, and the recommendation not to use echinocandin monotherapy if other options are available [12,25]. Even after accounting for the fact that the guidelines would only have become available partway through the pandemic, neither of the trusts studied adheres to this. Secondly, and importantly for both our trusts, the same pattern of antifungal prescribing that was seen prior to the COVID-19 pandemic continued during the COVID-19 pandemic, despite the complete change in aetiology of fungal infection. This suggests a strong influence of “habitual prescribing” and may reflect a lack of knowledge or understanding of the guidelines, a lack of familiarity with different antifungal medications, or fear of challenging other members of the team who are more senior or perceived to have more knowledge and experience. Current literature suggests that antimicrobial prescribing behaviours are more heavily influenced by peers and those considered leaders than by policies or guidelines [26,27,28]. In turn, the practices of junior colleagues are significantly influenced by their seniors. These influences are likely to be even more prevalent in times of great uncertainty such as during the COVID-19 pandemic. More research is needed into the reasons for non-adherence to guideline prescribing to improve the quality of interventions in this area. Antifungal stewardship has been shown to be key in improving antifungal prescribing behaviours but requires time, resources, and expertise [29].

After analysing antifungal use during COVID-19, it was determined that RBH showed high rates of antifungal prescription in COVID-19 patients. A systematic review suggests that the incidence of CAPA ranged between 2.5 and 35% [30]; however, 59% of patients at RBH received at least one course of antifungals. Accounting for potential diagnostic criteria for fungal disease, of those who received antifungals, 74% were deemed to have received empirical antifungal treatment. The actual figure is likely much higher, as we were reliant on non-bronchoalveolar lavage samples due to the impracticality of conducting bronchoalveolar lavage during the pandemic, and therefore, positive cultures may represent colonisation as opposed to a true disease. Despite the high use of antifungals, galactomannan testing was only performed in a minority of patients (18% of COVID-19 patients at RBH). Although serum galactomannan cannot be used to prove or exclude diagnosis of invasive aspergillosis, it can be considered a marker of mycological evidence of *Aspergillus* and as such forms part of the diagnostic guidelines, particularly when more invasive testing is not possible [12]. In contrast to RBH, at KCH, only 4% of COVID-19 patients received antifungals. This may in part be explained by the relative sizes of the intensive care departments as a proportion of all COVID-19 patients (KCH has a much larger bed base dedicated to non-intensive care patients); however, the heterogeneity of the use of antifungals between two trusts with site proximity alongside the high level of empirical use is further suggestive of general poor knowledge, understanding, and application of international guideline recommendations; a lack of familiarity with fungal diagnostics; and likely a fear of undertreating during a period of high uncertainty.

Given the global emergence of azole antifungal resistance, to understand the possible implication on antifungal prescription, we analysed azole-resistant *Aspergillus* cultures over time and identified an increase over time to a peak in 2019, in keeping with national and worldwide data [6]. The lower resistance rates during COVID-19 are likely explained by a reduction in the admission rates of patients with chronic lung disease such as cystic fibrosis or chronic pulmonary aspergillosis, leading to a reduction in samples being sent for analysis. CFTR modulators have also been shown to reduce *Aspergillus* colonisation, which may in turn impact the rates of resistance [31]. The relatively low levels of azole resistance seen both prior to and during the COVID-19 pandemic would suggest that this is not a direct explanation for the high levels of amphotericin usage seen.

In summary, our study findings highlight significant variability in antifungal use between two large NHS trusts with geographic proximity. Although partly accounted for by the differing populations served, the results during the COVID-19 pandemic demonstrate discrepancies, even when patients should have been receiving equitable care in accordance with national guidelines. We hypothesise that habitual prescribing, lack of knowledge or understanding of guidelines, an inadequate understanding of fungal diagnostics, and a fear of missed diagnoses may all be factors contributing to this variability. As is the case with antibiotics, inappropriate use or overuse of antifungals is harmful and associated with poorer outcomes [21,23], yet despite this, antifungal stewardship is significantly neglected in comparison. Work has begun to identify potential areas that could be targeted by future antifungal stewardship programmes, such as timely prescription review and de-escalation of therapy if invasive fungal disease is not proven [32], as well as into how such programmes could be structured and implemented [33]. However, much more work is needed in these areas, as well as in the development of point-of-care assays that could be used alongside other holistic interventions to improve stewardship. Prospective studies are needed to evaluate behaviours and factors that influence antifungal prescribing practice, and then to evaluate the effectiveness of interventions to increase adherence to guidelines. Interventions need to assess the local context of antifungal prescribing to ensure that such programmes are specific to the local population and needs of the trust.

Our study has several limitations. Due to the challenges in retrospective data acquisition within large, unstructured health records, we were unable to evaluate certain features that would have been interesting for further sub-analysis. For example, we could not obtain medical co-morbidities, and therefore we were unable to stratify antifungal use by indication pre-pandemic. As such, we were unable to determine whether the antifungal was being used as prophylaxis or for empirical or targeted treatment pre-pandemic, or whether co-morbidities, allergies, side effects, or prior treatment failures played a role in drug choice. We could also not tell whether an antifungal was being used outside of treating fungal infections—for example, for Leishmaniasis treatment. A further limitation during the COVID-19 pandemic was the assumption that antifungal usage was related to COVID-19 infection, and although this is most likely, we were unable to account for unrelated use. Within the COVID-19 patients, we were also not able to ascertain whether they were being treated in an intensive care unit environment or not (due to the redesignation of many general wards to become intensive care units), or any associated mortality. We therefore cannot make any comment on the severity of illness in which antifungals were being used and whether this could account for any discrepancy between the two hospitals. Thirdly, our diagnostic criteria for categorising antifungal use in COVID-19 patients did not include additional clinical features usually used to infer the likelihood of probable fungal infection such as radiological features, clinical symptoms, or co-morbidities.

## 5. Conclusions

This retrospective study analysing antifungal prescribing trends at two geoproximate large hospitals before and during the COVID-19 pandemic has highlighted the current significant challenges to optimise antifungal use, with prescribing largely dictated by habit, a lack of adherence to international recommendations, and high rates of empirical non-diagnostic-based prescribing. Further research and more resources are required to understand the impact of antifungal stewardship on improving antifungal prescribing behaviours in this setting and the effects on outcome.

## Figures and Tables

**Figure 1 jof-10-00787-f001:**
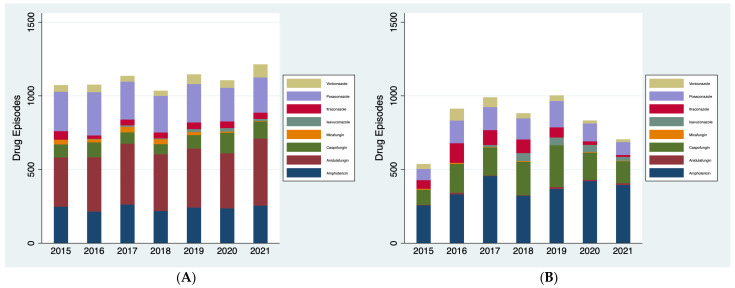
(**A**) Total systemic antifungal usage at KCH between January 2015 and December 2021. (**B**) Total systemic antifungal usage at RBH between January 2015 and December 2021.

**Figure 2 jof-10-00787-f002:**
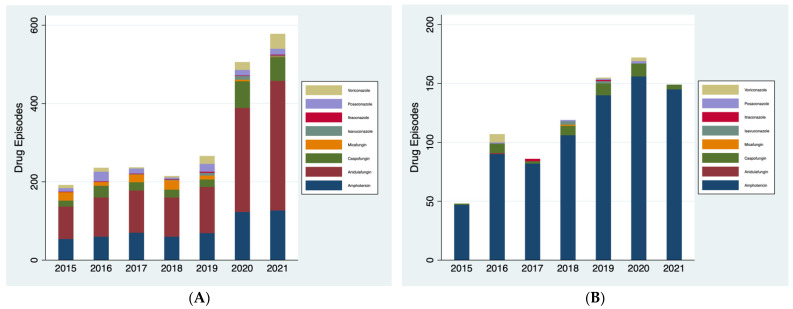
(**A**) Total systemic antifungal usage in the intensive care unit at KCH between January 2015 and December 2021. (**B**) Total systemic antifungal usage in the intensive care unit at RBH between January 2015 and December 2021.

**Figure 3 jof-10-00787-f003:**
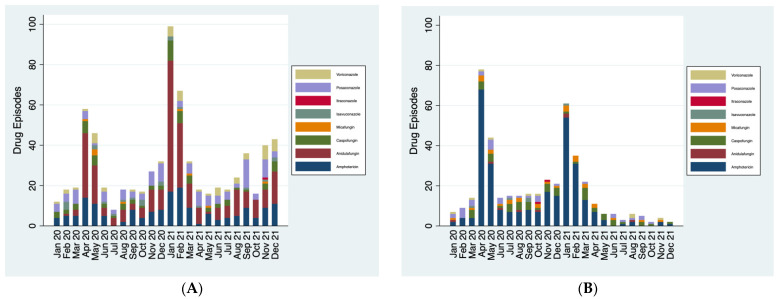
(**A**) Antifungal usage in patients with confirmed or suspected COVID-19 at KCH. (**B**) Antifungal usage in patients with confirmed or suspected COVID-19 at RBH.

**Figure 4 jof-10-00787-f004:**
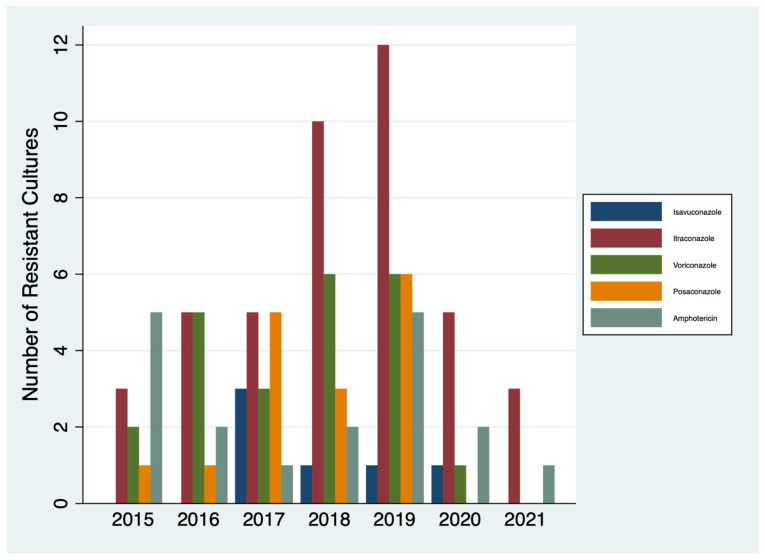
Number of azole-resistant *Aspergillus* species identified at RBH between 2015 and 2021.

**Table 1 jof-10-00787-t001:** Categorisation of antifungal use based on diagnostic testing in patients with confirmed or suspected COVID-19 at RBH.

	Positive Diagnostic Testing	Negative/No Diagnostic Testing
**Administered antifungals**	68	197
**Not administered antifungals**	18	168

## Data Availability

The original contributions presented in the study are included in the article. Further inquiries can be directed to the corresponding author.

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
