# Peer review of "The Effects of COVID-19 on Antifungal Prescribing in the UK—Lessons to Learn"

_jof, 2024, doi:10.3390/jof10110787_

Round 1
Reviewer 1 Report
I have reviewed the manuscript submitted by Katharine Pates , Zhendan Shang , Rebeka Jabbar , Darius Armstrong-James , Silke Schelenz , Jimstan Periselneris , Rossella Arcucci , Anand Shah which evaluated the “The effects of COVID-19 on antifungal prescribing in the UK – lessons to learn”. The manuscript was written clearly and objectively. Another positive point: The authors mentioned the limitations they encountered during the conduct of the study.
However, I suggest some modifications:
line 32: keywords 2x AND remove the numbers.
Figure 1A and 1B. The figures are cut off, making it impossible to see what is on the x-axis of the graphs.
Line 158: Table 1. Start the sentence with a capital letter. Center the information in Table 1.
Lines 281-287: I would remove these lines to the end, before the conclusion."
Author Response
Review 1 comments (black). Response (red).
I have reviewed the manuscript submitted by Katharine Pates , Zhendan Shang , Rebeka Jabbar , Darius Armstrong-James , Silke Schelenz , Jimstan Periselneris , Rossella Arcucci , Anand Shah which evaluated the “The effects of COVID-19 on antifungal prescribing in the UK – lessons to learn”. The manuscript was written clearly and objectively. Another positive point: The authors mentioned the limitations they encountered during the conduct of the study. However, I suggest some modifications:
Many thanks for your comments.
Comment 1: line 32: keywords 2x AND remove the numbers.
Response: thank you for pointing this out. We have made these corrections
Comment 2: Figure 1A and 1B. The figures are cut off, making it impossible to see what is on the x-axis of the graphs.
Response: thank you for pointing this out. We have made these corrections
Comment 3: Line 158: Table 1. Start the sentence with a capital letter. Center the information in Table 1.
Response: thank you for pointing this out. We have made these corrections
Comment 4: Lines 281-287: I would remove these lines to the end, before the conclusion."
Response: Thank you for your suggestion however the lines in question directly respond to the previous sentences of the summary paragraph, therefore we found moving them to after the limitations and before the conclusion as suggested made it harder to follow. We have therefore decided not to make this change.
Reviewer 2 Report
A respectful greeting, I congratulate the authors of the paper, which highlights the heterogeneity of criteria in the use of antifungals during the pre-pandemic and post-pandemic periods inclusive, but which also generates great uncertainty about the indications used for the prescription of antifungals during pandemic, each institution has a different reality in its microbiology, but the differences are so marked in the adherence to guidelines that it is difficult to justify the use of antifungals. In reality, what reveals a situation is that the stewardship of antifungals is an increasingly important need, the authors clarify possible biases and draw attention to the diagnostic difficulties of not having optimal samples due to the impossibility of doing invasive studies during the pandemic. I believe that the article contributes to knowledge and raises very important questions. Important that should be resolved in subsequent studies
A respectful greeting, I congratulate the authors of the paper, which highlights the heterogeneity of criteria in the use of antifungals during the pre-pandemic and post-pandemic periods inclusive, but which also generates great uncertainty about the indications used for the prescription of antifungals during pandemic, each institution has a different reality in its microbiology, but the differences are so marked in the adherence to guidelines that it is difficult to justify the use of antifungals. In reality, what reveals a situation is that the stewardship of antifungals is an increasingly important need, the authors clarify possible biases and draw attention to the diagnostic difficulties of not having optimal samples due to the impossibility of doing invasive studies during the pandemic. I believe that the article contributes to knowledge and raises very important questions. Important that should be resolved in subsequent studies
Author Response
Review comments (black). Response (red).
A respectful greeting, I congratulate the authors of the paper, which highlights the heterogeneity of criteria in the use of antifungals during the pre-pandemic and post-pandemic periods inclusive, but which also generates great uncertainty about the indications used for the prescription of antifungals during pandemic, each institution has a different reality in its microbiology, but the differences are so marked in the adherence to guidelines that it is difficult to justify the use of antifungals. In reality, what reveals a situation is that the stewardship of antifungals is an increasingly important need, the authors clarify possible biases and draw attention to the diagnostic difficulties of not having optimal samples due to the impossibility of doing invasive studies during the pandemic. I believe that the article contributes to knowledge and raises very important questions. Important that should be resolved in subsequent studies
Many thanks for your review and positive feedback on our study.
Reviewer 3 Report
An important study in terms of outlining how doctors prescribe, augmented by the availability of longitudinal data. The major message here is that guidelines are not followed, even in times of emergency. Two minor suggestions that could augment the manuscript's conclusions, although if they cannot be satisfied it is understandable, and they should be just mentioned in the limitations section.
1. There are no data on survival, particularly for covid-19 patients it would be useful both as a marker of efficacy AND severity
2. For covid-19 patients again, data on ICU admission and antifungal administration there, would be useful, particularly if it could explain the differences between the two hospitals.
3. Amphotericin use is not limited to fungal infections. Although obviously there is nit a lot of leishmaniasia treated in RBH, it should be simply mentioned in the text.
As described above, the comments are of general nature.
Author Response
Reviewer comments (black). Response (red).
An important study in terms of outlining how doctors prescribe, augmented by the availability of longitudinal data. The major message here is that guidelines are not followed, even in times of emergency. Two minor suggestions that could augment the manuscript's conclusions, although if they cannot be satisfied it is understandable, and they should be just mentioned in the limitations section.
Comments:
- There are no data on survival, particularly for covid-19 patients it would be useful both as a marker of efficacy AND severity.
- For covid-19 patients again, data on ICU admission and antifungal administration there, would be useful, particularly if it could explain the differences between the two hospitals.
- Amphotericin use is not limited to fungal infections. Although obviously there is nit a lot of leishmaniasia treated in RBH, it should be simply mentioned in the text.
Response: Thank you for your comments. We agree with all of the above as ways in which the data would have been enhanced, however unfortunately as you have summised it was not possible for us to obtain this data given the structure of the records. We have therefore added all the above to the limitations as suggested. Changes can be found in lines 295-307 of the manuscript.